# Beyond Cropped Regions: New Benchmark and Corresponding Baseline for Chinese Scene Text Retrieval in Diverse Layouts

Gengluo Li [1 2]   Huawen Shen [1 2]   Yu Zhou [3]

## Abstract

Chinese scene text retrieval is a practical task that aims to search for images containing visual instances of a Chinese query text. This task is extremely challenging because Chinese text often features complex and diverse layouts in real-world scenes. Current efforts tend to inherit the solution for English scene text retrieval, failing to achieve satisfactory performance. In this paper, we establish a **D**iversified **L**ayout benchmark for **C**hinese **S**treet **V**iew **T**ext **R**etrieval (**DL-CSVTR**), which is specifically designed to evaluate retrieval performance across various text layouts, including vertical, cross-line, and partial alignments. To address the limitations in existing methods, we propose **C**hinese **S**cene **T**ext **R**etrieval **CLIP** (**CSTR-CLIP**), a novel model that integrates global visual information with multi-granularity alignment training. CSTR-CLIP applies a two-stage training process to overcome previous limitations, such as the exclusion of visual features outside the text region and reliance on single-granularity alignment, thereby enabling the model to effectively handle diverse text layouts. Experiments on existing benchmark show that CSTR-CLIP outperforms the previous state-of-the-art model by 18.82% accuracy and also provides faster inference speed. Further analysis on DL-CSVTR confirms the superior performance of CSTR-CLIP in handling various text layouts. The dataset and code will be publicly available to facilitate research in Chinese scene text retrieval.

[1]Institute of Information Engineering, Chinese Academy of Sciences, Beijing, China [2]School of Cyber Security, University of Chinese Academy of Sciences, Beijing, China [3]VCIP & TMCC & DISSec, College of Computer Science, Nankai University, Tianjin, China. Correspondence to: Yu Zhou <yzhou@nankai.edu.cn>.

*Proceedings of the 42nd International Conference on Machine Learning*, Vancouver, Canada. PMLR 267, 2025. Copyright 2025 by the author(s).

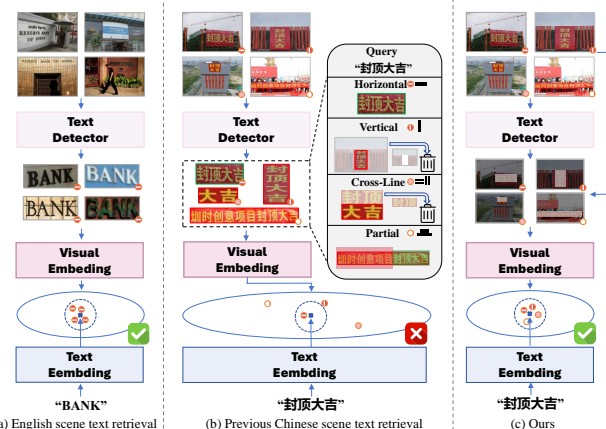

*Figure 1.* Pipeline comparison between (a) English scene text retrieval, (b) Chinese scene text retrieval adopted in previous work, and (c) Our proposed Chinese scene text retrieval framework that is based on full image information and multi-granularity alignment. The patterns in the circles represent the text layout forms.

## 1. Introduction

Text is an important object in scene images, and scene text related research topics including detection (Wang et al., 2022; Cao et al., 2025), recognition (Yang et al., 2025; Zhang et al., 2025), spotting (Lyu et al., 2025a; Wei et al., 2022), understanding (Shen et al., 2025; Zeng et al., 2023) and processing (Zeng et al., 2024b; Shu et al., 2025) have drawn increasing attention in recent years. Among them, scene text retrieval is an important topic of information retrieval, which involves searching for images that contain visual instances of a given query text within a collection of natural images (Mishra et al., 2013; Mafla et al., 2020). As texts in images generally convey valuable information, this task has been widely used in many applications, such as multimedia content retrieval, product recommendation, and automatic navigation (Karaoglu et al., 2016; Bai et al., 2018; Song et al., 2019).

In recent years, scene text retrieval has witnessed tremendous progress, while most of the existing methods focus on the language patterns of English. Whether these methods can be transferred to other language scenarios (especially

non-latin scripts) has not been fully explored. More specifically, as one of the most widely used non-latin languages, Chinese scene text retrieval differs distinctly from English scene text retrieval. As shown in Figure 1(a), in English setting, the query term is typically a single word, and text in images exhibits clear separations. As such, English scene text retrieval is essentially a simple local matching problem, which can be solved by measuring the similarity between the detected region and the query word. In contrast, in Chinese setting, there are no separations among words of the same sentences, the query terms can be any combination of consecutive characters, and Chinese has more highly variable layouts in real scenes. As shown in Figure 1(b), detection results often do not completely match the query words. Therefore, it is difficult to adapt the scene text retrieval pipeline from English to Chinese straightforwardly.

Targeting at the Chinese scene text retrieval task, Wang et al. (Wang et al., 2021) establish the CSVTR benchmark, and several studies have demonstrated promising performance on this benchmark. However, it can not well reflect the retrieval capabilities of models in real world. The query terms in this benchmark predominantly appear independently and are primarily horizontally oriented, neglecting the characteristics of diverse text layouts in Chinese scene text retrieval. In addition, most existing Chinese scene text retrieval models adopt paradigms developed for English scenes, utilizing cropped text regions from text detection results to represent the visual information of images, which brings several limitations. On the one hand, this method inevitably results in the loss of context, that is, the global semantic information can not contribute to the retrieval process, and parts of the query term may be lost. On the other hand, since redundant elements may be included in the detected boxes, the model is easily disturbed by irrelevant content. Moreover, previous methods strictly align the features of a text region with all the text it contains during training. This single-granularity alignment manner will undoubtedly cause some ambiguity, impairing models' ability to perceive and distinguish text. In Chinese scene text, the diverse combinations and layout forms of characters present significant challenges to previous solutions based on cropped regions. Therefore, enhancing the ability of scene text retrieval models to handle varied layouts is crucial for advancing scene text retrieval.

Since previous studies have not addressed the specific challenges of Chinese text retrieval, we propose a new benchmark and a scene text retrieval model that extends beyond cropped regions. As a first contribution, we introduce DL-CSVTR, a new **D**iversified **L**ayout benchmark for **C**hinese **S**treet **V**iew **T**ext **R**etrieval. We collect scene images containing visual instances of query terms in three of the most common text layouts, in addition to the horizontal layout, including (i) **vertical** layouts where the query terms are arranged vertically, (ii) **cross-line** layouts where query terms

span rows or columns, and (iii) **partial** layouts where query terms are connected to other text. DL-CSVTR combines diverse layouts to evaluate the ability of models to handle various text instances, simulating the requirements in real Chinese scene text retrieval applications.

Following it, we are committed to enhancing model performance on diverse text layouts by freeing the model from the limitations imposed by cropped regions on its visual perception range, which we believe is key to addressing the challenges of these layouts. To this end, we take advantages of the Contrastive Language-Image Pre-training (CLIP) model to retain full-image information, and convert the problem from text regions matching to global semantics enhanced layout patterns learning, as shown in Figure 1(c). Specifically, our proposed **C**hinese **S**cene **T**ext **R**etrieval **CLIP** model adopts a two-stage training paradigm. The goal of the first stage is to teach CLIP to focus on specific areas and enhance its OCR capabilities. During it, triplet inputs including the original image, the text segmentation map and the corresponding text are constructed, which are employed to train CLIP with a single-granularity alignment method. In the second stage, the model goes further into multi-granularity alignment, so as to learn the ability of perceiving diverse layouts. We apply the random granularity alignment algorithm to process the segmentation map and text in the triplets, producing many layout patterns for the model to learn. Meanwhile, global image features derived from the original CLIP model are injected in multi-scale layers of CSTR-CLIP, enabling the model to perform retrieval under the guidance of context.

Experiments on the previous Chinese scene text retrieval benchmark CSVTR show that CSTR-CLIP improves retrieval accuracy by 18.82% over the best previous method, while also delivering faster inference speed. Furthermore, both quantitative and qualitative experiments on DL-CSVTR confirm the effectiveness and generalization ability of CSTR-CLIP in handling diverse text layouts.

In summary, the contributions of this paper include:

- We conduct a thorough analysis of previous benchmark datasets and methods for Chinese scene text retrieval, identifying a significant gap in their ability to handle diverse text layouts. To address it, we propose a new benchmark, DL-CSVTR, which includes Chinese text image data for various layouts, to evaluate models' retrieval capabilities in real scenarios.

- We introduce CSTR-CLIP, a novel paradigm for Chinese scene text retrieval. CSTR-CLIP moves beyond the previous approach of text regions cropping, expands the model's visual receptive field by retaining full-image information, and enhances perception flexibility through multi-granularity alignment.

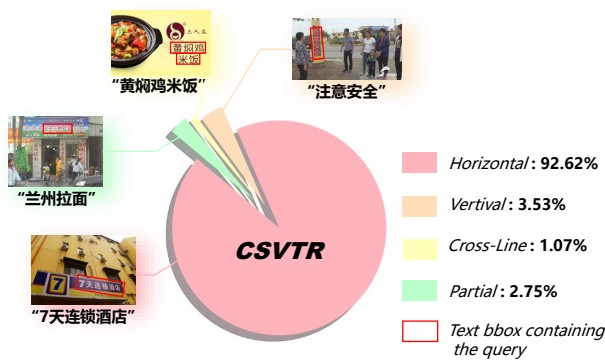

*Figure 2.* Layout distribution of the visual instance of the query word in the image in CSVTR.

- Our experiments demonstrate that the proposed method not only achieves state-of-the-art performance on the existing CSVTR dataset but also surpasses previous models in retrieval capabilities on the newly introduced DL-CSVTR benchmark.

## 2. Related Work

**Scene Text Retrieval Benchmark.** Mishra et al. (Mishra et al., 2013) is the first work to introduce the scene text retrieval task with the IIIT Scene Text Retrieval benchmark, which comprises numerous scene images and predefined query terms linked to specific images. The model accuracy is evaluated using Mean Average Precision (mAP), and speed is measured with Frames Per Second (FPS). Later researchers enhance this by using the Street View Text and TotalText datasets, creating richer benchmarks with text annotations as query terms (Wang & Belongie, 2010; Ch'ng & Chan, 2017). To examine the retrieval problem in other language setting, Wang et al. (Wang et al., 2021) introduce the CSVTR dataset for Chinese scene text retrieval, where query terms typically appear independently and are arranged horizontally, as shown in Figure 8. It ignores the challenges that the diverse text layout of Chinese brings to the scene text retrieval task.

**Scene Text Retrieval.** For scene text retrieval, a straightforward solution is to use text detection (Shu et al., 2023) and recognition(Qiao et al., 2020b; 2021) or end-to-end spotting (Lyu et al., 2025b) models to extract characters or words from images, simplifying the problem to traditional text retrieval (Mishra et al., 2013; Jaderberg et al., 2016; Liu et al., 2020; Qiao et al., 2020a; Liao et al., 2020). However, this approach often leads to error accumulation and limited performance (Huang et al., 2024). Some studies design manual text embeddings to convert characters into vector representations for improved retrieval robustness (Almazán et al., 2014; Ghosh et al., 2015; Ghosh & Valveny,

2015; Wilkinson & Brun, 2016; Gómez et al., 2018; Mafla et al., 2021), while they still suffer from the suboptimality of manually designed embeddings (Zhou et al., 2022b). Subsequently, cross-modal embeddings are proposed to unify visual and textual representations within a common feature space, achieving strong retrieval performance and speed (Gómez et al., 2017; Mhiri et al., 2019; Wang et al., 2021; Zeng et al., 2024a). Recently, visual embedding methods propose to transform query words into visual representations for retrieval via visual matching (Wen et al., 2023; Luo et al., 2024), achieving remarkable performance but at the cost of slower computational speeds. All these approaches rely on text detection to crop text regions for feature extraction. Although cropping enhances the visual features of the text region, the loss of semantic information outside this region has limited further improvements in scene text retrieval.

**CLIP's Attention Guidance and OCR Capability.** CLIP (Radford et al., 2021) is a vision-language model trained on large-scale data, possessing powerful representation capabilities (Materzyńska et al., 2022; Yu et al., 2023). Moreover, researchers find that many high-similarity image-text pairs in the LAION-2B dataset (Schuhmann et al., 2022) include visual instances of captions within the images (Lin et al., 2024) and synthetic text images can be utilized as visual prompts to enhance image classification performance (Li et al., 2022; Shi & Yang, 2023). These studies suggest that CLIP possesses potential OCR functionality and is well-suited for retrieval tasks (Luo et al., 2022; Baldrati et al., 2023; Saito et al., 2023). However, CLIP's perception often involves all image elements, resulting in a bias towards prominent objects (Xing et al., 2023). To enhance region awareness, methods like ReCLIP (Subramanian et al., 2022) crop images using bounding boxes, though this can lead to the loss of visual information. Red-Circle (Shtedritski et al., 2023) adds contours to guide CLIP's attention, while MaskCLIP (Zhou et al., 2022a) and AlphaClip (Sun et al., 2024) use masks to focus on specific regions, guiding CLIP to focus on target areas while preserving global context. Inspired by these works, our approach leverages CLIP to extract full-image information and facilitate scene text retrieval with CLIP's intrinsic text knowledge. Additionally, text location information is exploited to guide the model's perception area.

## 3. DL-CSVTR Benchmark

Based on our findings, we propose a **D**iversified **L**ayout benchmark for **C**hinese **S**treet **V**iew **T**ext **R**etrieval (**DL-CSVTR**), which considers the diverse text layouts of query terms' visual instances in Chinese scene text retrieval. Specifically, it includes data from three common scene text layout types, where the visual representation of query terms strictly adheres to their respective layouts: vertical, cross-

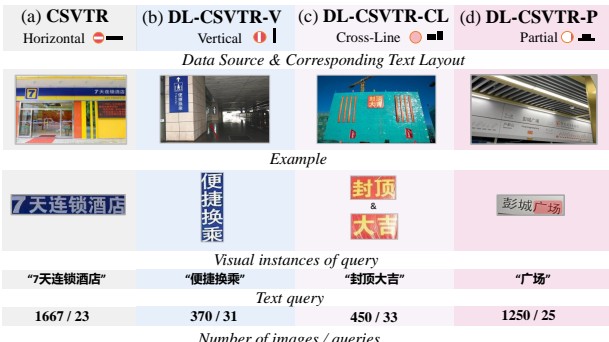

Figure 3. Common Chinese text layouts, where the visual representation of query terms is based on the text detection model's cropped results. In category (d), the red-highlighted area indicates the corresponding query term.

line, and partial, as shown in Figure 3.

To ensure the quality of annotations and minimize the impact of individual annotator bias, we adopt the strategy of simultaneous data collection by three annotators. First, we define 89 query terms for the three layouts. These query terms are primarily common conceptual nouns, including but not limited to trademark names, building names, and idioms. The details of the specific query word settings can be found in the supplementary material. Subsequently, the three annotators use these predefined query terms to search for images containing the specified text layouts of the query terms' visual instances. The results from all three annotators are then combined. After combining the results, the annotators screen the image set to remove duplicates and images where the query terms' visual representation did not match the specified text layouts, ensuring the uniqueness of the text layout for each query term in the images. The specific descriptions of each text layout are as follows:

**Vertical.** Vertical text is common in real-world scenes, characterized by text arranged from top to bottom, as shown in Figure 3(b). We collect 370 street view images containing vertically arranged text with 31 unique query terms. All these queries are visually represented in a vertical layout within the images.

**Cross-Line.** Cross-line text layouts occur when a conceptual noun spans multiple rows or columns, as shown in Figure 3(c). We collect 450 street view images featuring cross-line text layouts with 33 unique query terms. These queries are visually represented in either cross-row or cross-column layouts within the scene images.

**Partial.** Unlike English words, Chinese characters typically do not have distinct separations. When query terms appear in a long sentence, text line detection often includes characters beyond the query terms, as shown in Figure 3(d). We

collect 1,250 images with 25 query terms where the text detector's detected regions might include unrelated extra not unrelated characters.

We observe that different Chinese text layouts lead to varying granularities in the outputs of text detectors. For example, a cross-line layout may produce detection boxes that capture only portions of the query terms, while a partial layout might include additional characters. Besides, vertically arranged text is often associated with surrounding visual elements, such as nearby buildings or scenery. Previous scene text retrieval models typically crop text regions based on detector results, discarding all visual information outside the detection boxes and thereby losing valuable context. Additionally, these models employ single-granularity alignment, strictly aligning the visual features of a text region to the textual features of all text contained within, thereby limiting the model's broader perceptual scope. Consequently, our proposed DL-CSVTR benchmark introduces new challenges to assess the model's ability of handling diverse text layouts, with the data being used exclusively for testing purposes. Detailed settings of the query terms and image visualizations can be found in the supplementart material.

## 4. Method

This section introduces our proposed scene text retrieval model, **C**hinese **S**cene **T**ext **R**etrieval **CLIP** (**CSTR-CLIP**), which leverages multi-granularity perception guided by text regions and full-image information understanding capabilities. Additionally, it covers the training data-based random granularity alignment algorithm and explains how the model is applied to downstream tasks in scene text retrieval.

To overcome the limitations of the text region cropping paradigm, we use the entire image for feature extraction, expanding the visual receptive field. To distinguish different text labels in the image, we synthesize segmentation maps by annotating the position information of the text labels, using them to guide CLIP's focus on specific regions. A two-stage training process is designed to gradually equip CLIP with the ability to handle scene text retrieval across diverse text layouts. The model structure is illustrated in Figure 4.

**Stage 1: Training CLIP's OCR and regional perception capabilities.** As shown in Figure 4(a), the model is initialized with pre-trained Chinese CLIP weights (Yang et al., 2023), and Text Position Convolution is introduced to enhance the regional perception capabilities. During training, we keep the text encoder $E_T$ frozen.

Given an input image $I \in \mathbb{R}^{H \times W \times 3}$, it is first embedded with the CLIP RGB Convolutional layer $Conv_{RGB}$. Then, we generate a segmentation map $G \in \mathbb{R}^{H \times W \times 1}$ for each text label in $I$, highlighting the pixels in the text area while

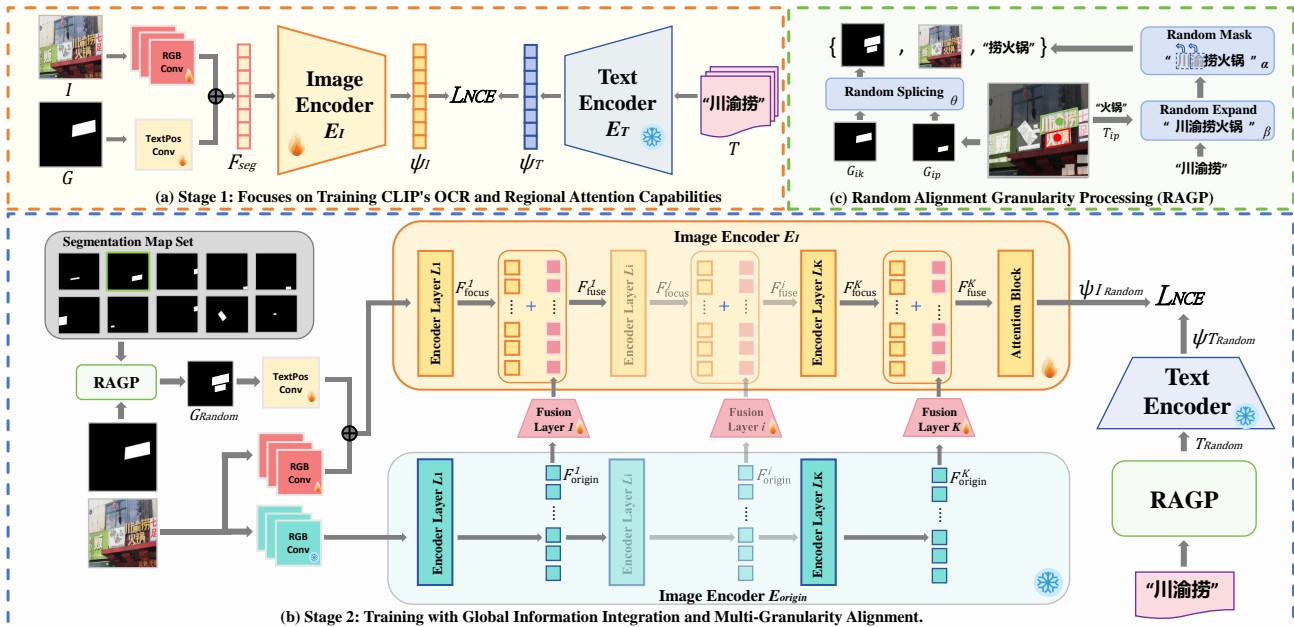

Figure 4. Two-stage training framework for the CSTR-CLIP model, including: (a) Training of model's region attention and OCR capabilities based on single-granularity alignment; (b) Adapting CSTR-CLIP to diverse text layouts using random multi-granularity alignment and integrating global features. (c) The Random Alignment Granularity Processing (RAGP) module in stage 2.

setting the background pixels to zero. We design a single-channel convolutional layer, $Conv_{Textpos}$, that accepts $G$ as input to obtain the guided embedding. After that, these two embeddings are fused:

$$F_{\text{seg}} = Conv_{RGB}(I) + Conv_{Textpos}(G) \qquad (1)$$

Next, $F_{\text{seg}}$ is fed into the CLIP image encoder $E_I$ to obtain the image feature $\psi_I \in \mathbb{R}^d$, where $d$ is the dimension of the CLIP embedding space. The corresponding text is fed into the text encoder $E_T$ to obtain the text feature $\psi_T \in \mathbb{R}^d$:

$$\psi_I, \psi_T = E_I(F_{\text{seg}}), E_T(T) \qquad (2)$$

During training, for each image $I_i$, we generate triplets $\{I_i, G_{ik}, T_{ik}\}$ by iterating over its text box annotations, where $i$ is the image index and $k$ is the text annotation index. The number of text line annotations determines the number of triplets generated. We use all triplets generated from the images as the training set, apply a single-granularity alignment method to align text regions with their contents, and train the model using NCE Loss:

$$L_{\text{NCE}} = -\frac{1}{N} \sum_{i=1}^{N} \log \frac{\exp(\psi_I^i \cdot \psi_T^i / \tau)}{\sum_{j=1}^{N} \exp(\psi_I^i \cdot \psi_T^j / \tau)} \qquad (3)$$

where $\tau$ is the temperature parameter.

This stage enhances CLIP's OCR capabilities and its perception guided by the specified regions. The model is initially trained with synthetic data and then fine-tuned using real-world scene data. It can direct perception to specified regions through $Conv_{Textpos}$. However, the single-granularity alignment method used during training leads to losing some visual features in non-guided areas.

**Stage 2: Training with Global Information Integration and Multi-Granularity Alignment.** We utilize the frozen original Chinese CLIP visual encoder $E_{\text{origin}}$ to enhance our first stage model, as illustrated in Figure 4(b). The scene image $I$ is also fed into $E_{\text{origin}}$ simultaneously. For each layer $l$, the intermediate features $F_{\text{focus}}^l$ from $E_I$ are augmented by the corresponding global features $F_{\text{origin}}^l$ from $E_{\text{origin}}$:

$$F_{\text{fuse}}^l = F_{\text{focus}}^l + \text{FusionLayer}(F_{\text{origin}}^l) \qquad (4)$$

where the Fusion Layer is a 1x1 convolutional layer.

This process aims to recover the visual feature loss in non-perception regions from the first stage, thereby enhancing the model's understanding of global information. This stage exclusively uses real-world scene data for training.

**Random Alignment Granularity Processing (RAGP).** To augment the training data for multi-granularity alignment, we apply RAGP on segmentation maps and text inputs, as illustrated in Figure 4(c). According to our previous analysis, single-granularity alignment limits the model's perception to

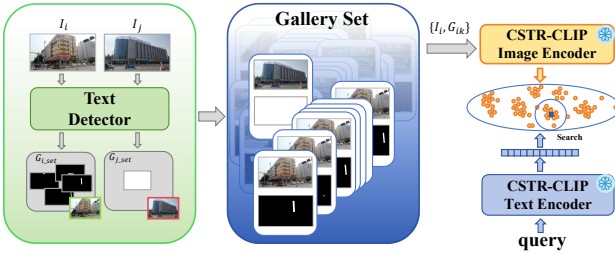

*Figure 5.* The retrieval pipeline of CSTR-CLIP.

the entire text within the guided region, thereby reducing its flexibility. In cross-line and partial layouts, the corresponding text region may miss some queries or include redundant text, making single-granularity alignment less than ideal. The key to solving this problem is to enable the model to flexibly perceive text elements inside and near the text region, disrupting the exact match between the segmentation map $G_{ik}$ and the corresponding text $T_{ik}$ during training.

We truncate $T_{ik}$ by masking $m$ characters from both ends, ensuring a minimum length of 2, with the deletion probability controlled by a hyperparameter $\beta$. We assume that semantically related text elements are spatially close in the image. For $T_{ik}$, we identify the text $T_{ip}$ whose bounding box centroid is closest to $T_{ik}$ and extend it in the direction connecting their centroids, with the probability of adding text controlled by a hyperparameter $\alpha$. For the segmentation map $G_{ik}$, we locate the segmentation map $G_{ip}$ whose bounding box centroid is closest to $T_{ik}$ and concatenate them, with the probability of concatenation controlled by a hyperparameter $\theta$. In general, the text is first randomly expanded and then randomly masked, and the processing of the segmentation map is independent of the text processing.

**CSTR-CLIP for Inference.** As illustrated in Figure 5, we use PaddleOCR to detect text, and RAGP is not applied in the inference stage. Detected text regions are converted into segmentation maps and paired with the original image, forming the gallery set. If no text regions are detected, a fully highlighted image is used as the segmentation map. All related pairs in the gallery set are processed by the visual encoder to extract visual features, and the query word is encoded into text features by the text encoder. For an image, multiple binary groups are formed with the various text regions identified by the detector, resulting in multiple visual features guided by different text regions. The highest similarity score between these visual and text features determines the final score for the image's visual representation.

## 5. Experiments

**Training Datasets.** We follow the training setting of previous studies (Wen et al., 2023; Luo et al., 2024), employing

a combination of synthetic and real data for model training.

- **SynthText-CH.** Following SynthText-900K (Gupta et al., 2016), we creat the SynthText-CH dataset with 300K samples, recording bounding box coordinates to generate segmentation maps for the first training stage.

- **ReCTS.** The ReCTS (Zhang et al., 2019) dataset comprises annotated Chinese text images in natural scenes. Segmentation maps are generated from these annotations for both training stages. It serves as the data for the second training stage.

**Test Datasets.** The test data comprises the existing Chinese scene text retrieval benchmark CSVTR and our proposed DL-CSVTR.

- **CSVTR.** This dataset includes 1,667 images which are gathered using Google Image Search, corresponding to 23 predefined queries. The visual appearance of the query terms in the images is primarily in a horizontal layout and presented independently.

- **DL-CSVTR.** The DL-CSVTR dataset proposed in this paper includes three types of text layout benchmarks, with a total of 2,070 images and 89 predefined query terms. This dataset will be used to evaluate the model's retrieval ability across different text configurations. The tests will be conducted on the subsets corresponding to the three types of text layouts.

**Baselines.** We use previously published scene text retrieval models tested on CSVTR as baselines. We reproduce and evaluate the most competitive methods on DL-CSVTR. Since earlier methods do not utilize the CLIP pre-trained model, we replace the backbones in previous works (Wang et al., 2021; Luo et al., 2024) with the pre-trained CLIP model to ensure a fair comparison. These methods represent the state-of-the-art in cross-modal and visual embedding paradigms. The source of training data is the same as CSTR-CLIP, with the text area cropped according to the annotations for the training set.

**Implementation Details.** Due to CLIP's limited input size and the small text in scene images, we expand the input resolution of CSTR-CLIP to 640×640. Yet, this will lead to inappropriate positional encoding in the self-attention layer that processes the features extracted by visual extraction module, so we replace them with learnable ones, initialized using nearest neighbor interpolation. The visual extraction module utilizes the pre-trained ResNet50 version.

We optimize CSTR-CLIP using the Adam optimizer with an initial learning rate of 1e-6 and a batch size of 48. The first stage is trained for 8 epochs (6 with synthetic data and

2 with real data), while the second stage is trained for 10 epochs. In RAGP, we set the hyperparameters controlling the randomness to $\beta = 0.2$, $\alpha = 0.3$, and $\theta = 0.2$. Models are trained on an NVIDIA A6000 GPU and tested on a GTX 1080 GPU.

## 5.1. Performance on CSVTR benchmark

| Method | mAP % | FPS |
|---|---|---|
| Mishra et al(Mishra et al., 2013) | 4.79 | 0.10 |
| TDSL (Wang et al., 2021) | 60.23 | 12 |
| VSTR (Wen et al., 2023) | 63.17 | 11 |
| Luo et al(Luo et al., 2024) | 69.75 | 11.2 |
| CLIP (640x640)(Radford et al., 2021) | 57.32 | **47.6** |
| TDSL (Wang et al., 2021)[†] | 78.36 | 12.3 |
| Luo et al(Luo et al., 2024)[†] | 80.74 | 10.5 |
| CSTR-CLIP | **88.57** | 21.5 |

Table 1. Comparison with existing methods on CSVTR. [†] represents the version that replaces the backbone with the CLIP encoder. We highlight the **best** and the second results.

We compare our model with previous models on CSVTR, and the experimental results are shown in Table 1. The results clearly demonstrate that our model outperforms the previous methods, improving retrieval accuracy by 18.82% while maintaining competitive inference speed. Additionally, our method continues to excel even when the encoders of state-of-the-art models based on cross-modal and visual embeddings are replaced with the CLIP encoder as an additional baseline. We attribute this improvement to our method's ability to retain visual features outside the text regions from a global image perspective. In scenarios where the detector fails to detect the text regions or the text regions in the image have poor clarity, our full-image based method leverages rich visual context information to achieve superior retrieval capabilities.

## 5.2. Performance on DL-CSVTR benchmark

As shown in Table 2, CLIP demonstrates strong retrieval ability across all DL-CSVTR benchmarks, highlighting the importance of perceiving beyond cropped regions for understanding diverse text layouts. While methods based on the cropped regions paradigm surpass the CLIP's retrieval ability in vertical layouts, they perform poorly in cross-line and partial layouts. This indicates that the cropped regions paradigm struggles to address the challenges posed by these more complex configurations.

However, after single-granularity training in the first stage, our method surpasses previous methods based on the cropped regions paradigm across all DL-CSVTR benchmarks, and further surpasses all baselines to achieve the best retrieval performance after the second stage of train-

ing. For vertical layouts, our method outperforms cropped region-based methods by leveraging full-image information, demonstrating that understanding information beyond cropped regions is crucial for improving the model's understanding of vertical text layouts. In cross-line and partial layouts, the model, limited by single-granularity alignment, does not surpass the CLIP baseline after the first stage. However, in the second stage, the fusion of full-image information with multi-granularity perception training, guided by text regions, overcomes the limitations of single-granularity alignment and non-guided region information loss from the first stage, resulting in impressive performance.

## 5.3. Ablation Study

In this section, we conduct a series of ablation experiments based on our model training stages. The detailed results are presented in Table 3.

**Effectiveness of Stage 1 settings.** We conduct an ablation study on the Stage 1 settings to evaluate their effectiveness. (See #1, #2 and #3) When the visual encoder is frozen, $Conv_{Textpos}$ can be considered a data augmentation method, adapting the image to the visual encoder's attention bias through single-granularity alignment. This approach performs well on CSVTR, where query terms typically appear independently and in a horizontal layout, resulting in better performance due to accurate text detection. However, in the more diverse settings of DL-CSVTR, the performance declines due to suboptimal detector outputs, particularly in partial and cross-line layouts where only parts of the query terms or extraneous elements are captured.

Involving the visual encoder $E_I$ in model training enhances performance across all datasets (See #2 and #3). This improvement results from fine-tuning the OCR capabilities and enhancing $Conv_{Textpos}$'s guidance function at the visual encoder level. The gains are especially notable on CSVTR and DL-CSVTR-V, where jointly training the image encoder $E_I$ and $Conv_{Textpos}$ results in optimal attention guidance. However, improvements on DL-CSVTR-CL and DL-CSVTR-P are less pronounced, as the single-granularity alignment method overly restricts the model's focus, resulting in poorer performance when only part of the query's visual representation or irrelevant elements are included in the guidance area.

**Effectiveness of Stage 2 settings.** For the Stage 2, the ablation results of each setting are also reported (See #3, #4, #5 and #6). The inclusion of full-image features on CSVTR leads to some improvements, but RAGP does not yield significant gains, as query terms in CSVTR typically appear independently and in a horizontal layout. In DL-CSVTR-V, the addition of full-image information brings significant improvement, validating the importance of full-image layout information in understanding vertically aligned text. For

| Method | DL-CSVTR-V (mAP %) | DL-CSVTR-CL (mAP %) | DL-CSVTR-P (mAP %) |
|---|---|---|---|
| Luo et al(Luo et al., 2024) | 39.87 | 21.98 | 13.95 |
| CLIP (640x640) (Radford et al., 2021) | 55.48 | 54.43 | 37.31 |
| TDSL(Wang et al., 2021)[†] | 62.51 | 40.46 | 24.88 |
| Luo et al(Luo et al., 2024)[†] | 52.73 | 34.23 | 29.91 |
| CSTR-CLIP (Stage1) | 74.50 | 45.98 | 33.25 |
| CSTR-CLIP | **84.44** | **65.56** | **61.85** |

*Table 2.* Comparison with existing methods on DL-CSVTR. [†] represents the version that replaces the backbone with the CLIP encoder. We highlight the **best** and the second results.

| # | Stage1 $Conv_{Textpos}$ | IE | Stage2 Global features | RAGP | mAP(%) CSVTR | DL-CSVTR-V | DL-CSVTR-CL | DL-CSVTR-P |
|---|---|---|---|---|---|---|---|---|
| 1 | ✗ | ✗ | ✗ | ✗ | 57.32 | 55.48 | 54.43 | 37.30 |
| 2 | ✔ | ✗ | ✗ | ✗ | 59.50 | 53.17 | 42.15 | 30.70 |
| 3 | ✔ | ✔ | ✗ | ✗ | 86.25 | 74.50 | 45.98 | 33.25 |
| 4 | ✔ | ✔ | ✔ | ✗ | 88.41 | 83.91 | 57.82 | 43.19 |
| 5 | ✔ | ✔ | ✗ | ✔ | 86.31 | 74.01 | 60.08 | 52.51 |
| 6 | ✔ | ✔ | ✔ | ✔ | **88.57** | **84.44** | **65.56** | **61.85** |

*Table 3.* Ablation experimental results. In Stage 1, we perform ablation studies on the use of $Conv_{Textpos}$ and the inclusion of the image encoder (IE) in training. In Stage 2, we ablate the inclusion of global features from the original CLIP model and the use of multi-granularity alignment processing. We highlight the **best** and the second results.

DL-CSVTR-CL and DL-CSVTR-P, integrating full-image information and random granularity alignment processing significantly enhances the model's ability to retrieve text across rows and in partial layouts. RAGP refines the model's attention to partial text within the guided area and external text content, while full-image information enhances the model's overall perception and understanding of the image content. As shown in Figure 6, we visualize the fused features of the intermediate layers using the merged channel visualization method. The visualization results clearly indicate that after the Stage 2 training, CSTR-CLIP not only enhances its perception of the entire image but also improves its understanding of the text information near the guided area and further refines the internal text elements.

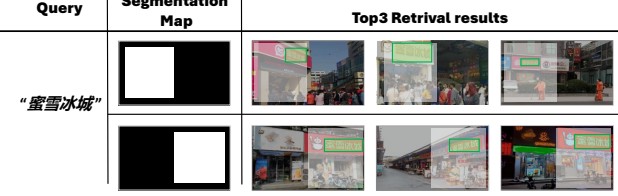

*Figure 7.* An example of region-specified scene text retrieval. The green boxes in the search results indicate visual instances of the query terms.

### 5.4. Interactive Region-Specified Scene Text Retrieval

Our approach paves the way for more diverse text retrieval methods, enabling region-specific retrieval. Specifically, $Conv_{Textpos}$ allows the model to focus on user-defined regions, while multi-granularity alignment enhances the flexibility of the model's perception, permitting users to custom-design segmentation maps. As illustrated in Figure 7, by highlighting specific parts of the segmentation map, we direct the model's attention to the corresponding image region, with the visual examples of the query terms in the top-ranked results mostly appearing in the guided region. This enables our model to perform more accurate scene text retrieval with region-specific guidance.

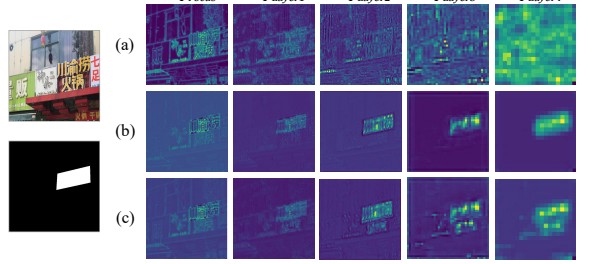

*Figure 6.* Visualization results of model intermediate layer features, including (a) original CLIP, (b) CSTR-CLIP after Stage 1 training, (c) CSTR-CLIP after Stage 2 training.

# 6. Conclusion

This paper identifies the limitations of the previous scene text retrieval paradigm in handling the diverse text layouts of Chinese. To tackle it, a new benchmark DL-CSVTR is proposed, which contains three types of common text layouts for evaluating models' retrieval capabilities in real scenarios. To overcome the limitations of the previous text region cropping paradigms in Chinese scene text retrieval, we propose the CSTR-CLIP model. Our model is designed with a multi-granularity-aware paradigm, leveraging the entire image while being guided by text regions. This approach not only achieves the best retrieval accuracy and speed on previous benchmarks but also demonstrates significant performance improvements under various text layout conditions on DL-CSVTR. The text region-guided design allows users to specify regions of interest for more detailed retrieval in practical applications.

# Acknowledge

Supported by the National Natural Science Foundation of China (Grant NO 62376266 and 62406318), Key Laboratory of Ethnic Language Intelligent Analysis and Security Governance of MOE, Minzu University of China, Beijing, China.

# Impact Statement

This paper presents work whose goal is to advance the field of Machine Learning. There are many potential societal consequences of our work, none which we feel must be specifically highlighted here.

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

## A. Analysis of DL-CSVTR

This section supplements the statistical analysis of CSVTR and the definition of query terms in our proposed DL-CSVTR.

### A.1. Analysis of text layout distribution in CSVTR

We have analyzed the four representations of query terms in CSVTR. The statistics are based on the image as the basic unit, with priority given to horizontal, then partial, then cross-line, and finally vertical layouts. For example, if the visual representation of a query term in an image exists both horizontally and cross-line, the image is classified as having a horizontal layout. The statistical results are shown in Figure 8.

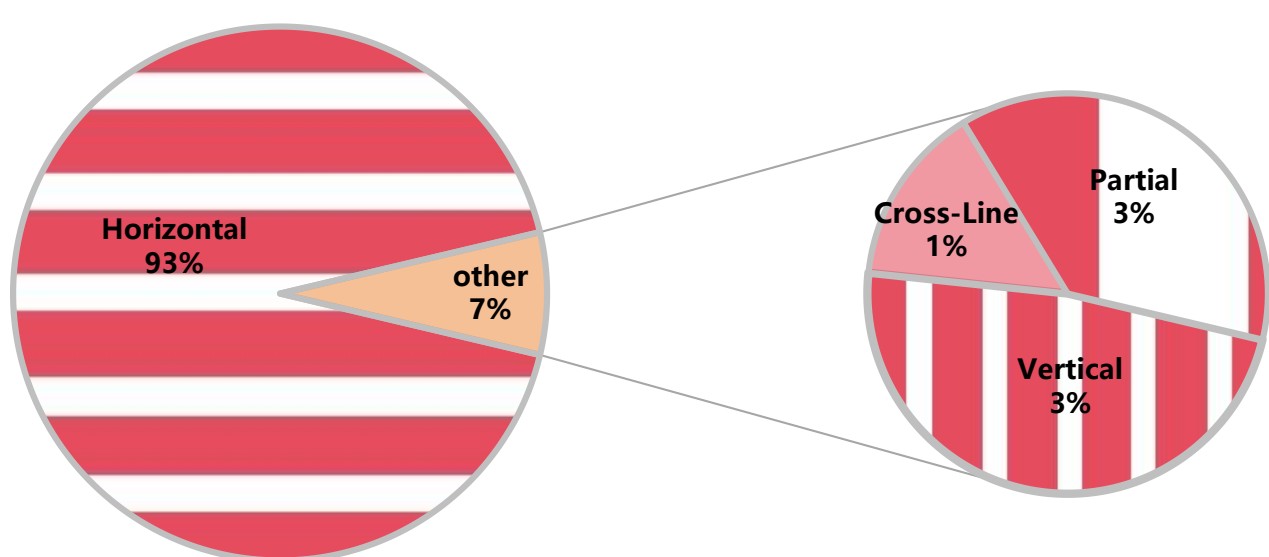

*Figure 8.* Distribution of text layouts for visual representations of query terms in CSVTR

The statistical results clearly show that the visual layout of most query terms is horizontal, which does not reflect the challenges that diverse text layouts pose to scene text retrieval tasks.

### A.2. Defining the query terms in DL-CSVTR

The query terms need to appear in a variety of scene images, so they must be common, high-frequency words. In the Chinese context, these are mostly Chinese trademark names, idioms, or noun concepts with specific semantic meanings. Based on our analysis and the definition of query terms in CSVTR, we designed query terms for the three types of text layout benchmarks in DL-CSVTR, as shown in Figure 2.

For each defined query term, we used common image search engines and manually screened images to find visual representations of the query terms that match the specific text layouts, ensuring that the three layouts remain distinct in our DL-CSVTR.

### A.3. Sample visualization in DL-CSVTR

We visualize data from the three benchmarks in DL-CSVTR to provide a clearer understanding of the DL-CSVTR dataset. The visualization results are presented in Figure 10.

In DL-CSVTR-V, the visual representation of the query terms is in a vertical layout, while in DL-CSVTR-CL and DL-CSVTR-P, the representation spans rows or columns and is connected to other characters.

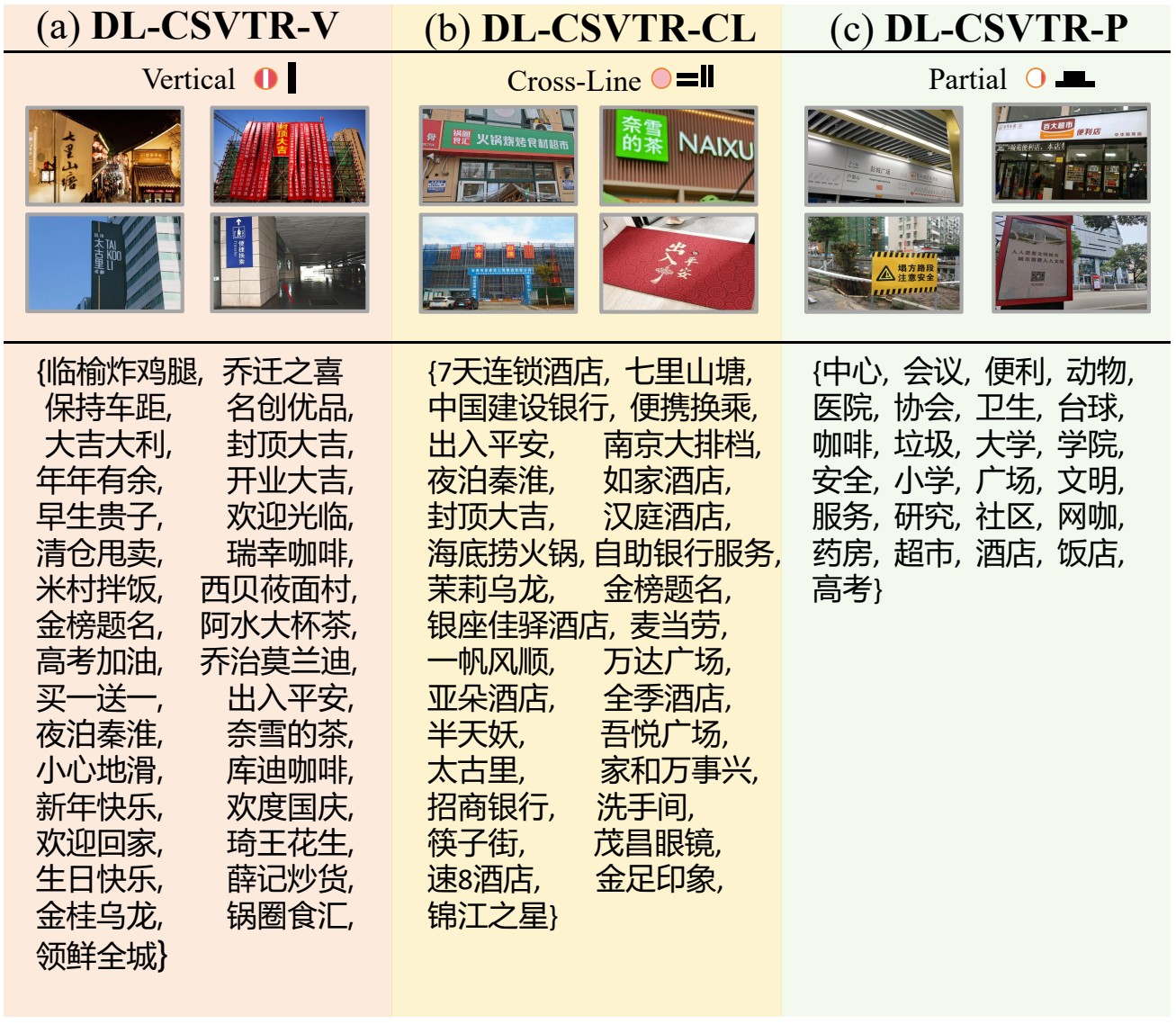

*Figure 9.* Distribution of text layouts for visual representations of query terms in CSVTR

## A.4. Supplementary examples of DL-CSVTR

We provide a portion of the DL-CSVTR dataset, ensuring that the images do not reveal information about the author or associated entities. The dataset includes the following:

1. DL-CSVTR-V: This folder contains a subset of vertically arranged data, where the visual instances of the query words appear in a vertical layout within the images.

2. DL-CSVTR-CL: This folder contains a subset of cross-row layout data, where the visual instances of the query words appear in a cross-row or cross-column layout within the images.

3. DL-CSVTR-P: This folder contains a subset of partial layout data, where the visual instances of the query words appear connected to other text within the images.

---

**Algorithm 1** Training Triplet Generation

---

**Input**: OCR dataset with images $\{I_i\}$ and their text annotations $\{T_{ik}\}$, where $i$ is the image index and $k$ is the text line index.

**Output**: Set of triplets $\{(I_i, G_{ik}, T_{ik})\}$ for training

 1: **for** each image $I_i$ in the dataset **do**
 2:     Initialize an empty set of triplets $\mathcal{T}$
 3:     **for** each text line annotation $T_{ik}$ in $I_i$ **do**
 4:         Initialize a grayscale image $G_{ik}$ of the same size as $I_i$ with all pixel values set to 0
 5:         Get the polygonal coordinates $P_{ik}$ of the text line $T_{ik}$
 6:         Set the pixel values within the polygon $P_{ik}$ in $G_{ik}$ to 255
 7:         Add the triplet $(I_i, G_{ik}, T_{ik})$ to $\mathcal{T}$
 8:     **end for**
 9: **end for**
10: Store the triplets $\mathcal{T}$
11: **return** All stored triplets $\{(I_i, G_{ik}, T_{ik})\}$

---

**Algorithm 2** Retrieval bigrams Generation for Multiple Images using PaddleOCR

---

**Input**: Set of original images $\{I_i\}$
**Output**: Set of retrieval pairs $\{(I_i, G_{ik})\}$

 1: Initialize an empty set of pairs $\mathcal{P}$
 2: **for** each image $I_i$ in the image set $\{I_i\}$ **do**
 3:     Use PaddleOCR to perform text detection on image $I_i$, obtaining a set of bounding boxes $\{B_{ik}\}$
 4:     **for** each bounding box $B_{ik}$ in detection results $\{B_{ik}\}$ **do**
 5:         Initialize a grayscale image $G_{ik}$ of the same size as $I_i$ with all pixel values set to 0
 6:         Get the polygonal coordinates $P_{ik}$ of the bounding box $B_{ik}$
 7:         Set the pixel values within the polygon $P_{ik}$ in $G_{ik}$ to 255
 8:         Add the pair $(I_i, G_{ik})$ to $\mathcal{P}$
 9:     **end for**
10: **end for**
11: **return** All generated pairs $\mathcal{P}$

---

# B. Details of the Model

This section provides additional explanations regarding the model details and specific implementation.

### B.1. Training triple generation

We generate training triplets based on the text line annotations of the images for both training and testing in the first and second stages. Specifically, the data generation algorithm creates the segmentation map mentioned in the text, based on the text line annotations. This segmentation map is a grayscale image that forms a triplet with the corresponding original image and text. Please refer to the pseudocode provided in Algorithm 1.

### B.2. Retrieval bigrams generation

In the retrieval stage, thanks to the standard two-tower model we use, we only need to create a segment map for the images in the gallery to get the bigrams. We use Paddle OCR, an OCR detection model with strong performance and fast inference speed, as the detector. The detector is used to give the location annotations of multiple text regions and generate a segmentmap to obtain candidate bigrams. Please refer to the pseudocode provided in Algorithm 2

### B.3. Supplementation of RAGP

To better understand the processing order of RAGP, we provide pseudocode to illustrate the process. Please refer to Algorithm 3.

---

**Algorithm 3** Random Granularity Alignment Processing

---

**Input**: Text inputs $T_{ik}$, Grayscale images $G_{ik}$
**Parameter**: $\beta, \alpha, \theta$
**Output**: Processed text inputs and grayscale images

1: **for** each text input $T_{ik}$ **do**
2:     **Random Expand:**
3:     Find $T_{ip}$ whose centroid is closest to $T_{ik}$
4:     With probability $\alpha$, merge $T_{ik}$ and $T_{ip}$
5:     **Random Mask:**
6:     **if** length($T_{ik}$) $\geq 4$ **then**
7:         With probability $\beta$, delete $m$ characters from $T_{ik}$, ensuring length $\geq 2$
8:     **end if**
9: **end for**
10: **for** each grayscale image $G_{ik}$ **do**
11:     **Random Splicing:**
12:     Find $G_{ip}$ whose centroid is closest to $G_{ik}$
13:     With probability $\theta$, merge $G_{ik}$ and $G_{ip}$
14: **end for**

---

The pseudocode clearly illustrates the process of the RAGP we proposed. In the actual code implementation, we accomplish this random processing algorithm using the getitem function and other custom-defined functions when creating the dataset.

## C. Visualization of Retrieval Results.

### C.1. Retrieval visualization on DL-CSVTR

To intuitively analyze the retrieval capabilities of STR-CLIP and previous methods on Chinese text retrieval with diverse layouts, visualization results for three different text layout benchmarks from DL-CSVTR are presented in Figure 11.

Under the DL-CSVTR-V benchmark, as shown in Figure 11(a), previous methods based on cropping text regions discard visual information outside the text region, hindering the model's ability to understand vertically arranged text. For instance, in positive examples, we observed that vertical text in the scene is often associated with surrounding visual elements, such as vertical billboards and architectural styles, which are crucial for the model's comprehension of vertical layouts. Our method, STR-CLIP, achieves the best retrieval results by understanding text information and related visual elements from a full-image perspective. The AP comparison for each query term is shown in the figure 12.

Under the DL-CSVTR-CL benchmark, as shown in Figure 11(b), our proposed model benefits from breaking the limitations of the cropped text region paradigm. Even if the text region that guides the model to perceive only contains part of the query words, our full-image information perception and multi-granularity perception guided by the text region can solve this problem well. The AP comparison for each query term is shown in the figure 13.

Under the DL-CSVTR-P benchmark, as shown in Figure 11(c), our proposed model outperforms the previous single-granularity matching paradigm by enhancing the perception of specific elements within the text region. The AP comparison for each query term is shown in the figure 14.

### C.2. Retrieval visualization on CSVTR

The AP comparison for each query term is shown in the figure 15.

## D. Supplementary of Experiments

### D.1. Details of baseline reproduction

We modified the open-source code of TDSL (Wang et al., 2021) and Luo et al (Luo et al., 2024) which represent the most advanced scene text retrieval models for cross-modal embedding and visual embedding, respectively. Specifically, for TDSL,

we replaced the language and visual encoders with the pre-trained Chinese CLIP encoder and fine-tuned it using cropped text regions and annotations. For VSTR, we replaced the dual-end visual encoder with the visual encoder from Chinese CLIP and trained it using cropped text regions and annotated text instance images generated from annotations. After replacing the encoder, both models demonstrated stronger retrieval ability on the previous CSVTR benchmark.

### D.2. Supplement of model code

We provide the model code and data processing scripts used in our work, along with some data to illustrate our model training process. We have ensured that all content potentially leaking personal information has been removed. Specifically, the provided code and related data include the following components:

1. **CSTR-CLIP-Stage1.py**: This file contains the model code and dataset loading script for the first stage of training CSTR-CLIP.

2. **CSTR-CLIP-Stage2.py**: This file contains the model code and dataset loading script for the second stage of training CSTR-CLIP. The RAGP algorithm we designed is incorporated into the dataset design.

3. **train-img**: This folder contains a portion of the data used for training, including original images and segmentation maps.

| # | Benchmark | Query | Example |
|---|---|---|---|
| **(a)** | **DL-CSVTR-V** | **"七里山塘"** |  |
| | | **"如家酒店"** |  |
| | | **"麦当劳"** |  |
| **(b)** | **DL-CSVTR-CL** | **"封顶大吉"** |  |
| | | **"库迪咖啡"** |  |
| | | **"西贝莜面村"** |  |
| **(c)** | **DL-CSVTR-P** | **"安全"** |  |
| | | **"超市"** |  |
| | | **"咖啡"** |  |

*Figure 10.* Sample presentation of three benchmarks in DL-CSVTR. Best viewed in zoom.

| # | Benchmark | Query | Method | Retrieval Results |
|---|---|---|---|---|
| (a) | DL-CSVTR-V | "锦江之星" | Visual Embeding |  |
| | | | Cross Model Embeding |  |
| | | | CSTR-CLIP |  |
| (b) | DL-CSVTR-CL | "锅圈食汇" | Visual Embeding |  |
| | | | Cross Model Embeding |  |
| | | | CSTR-CLIP |  |
| (c) | DL-CSVTR-P | "网咖" | Visual Embeding |  |
| | | | Cross Model Embeding |  |
| | | | CSTR-CLIP |  |

*Figure 11.* Visualization of retrieval results. (a) An example on DL-CSVTR-V benchmark, in which rank@1-5 retrieval results are provided. (b) An example on DL-CSVTR-CL benchmark, in which rank@1-5 retrieval results are provided. (c) An example on DL-CSVTR-P benchmark, in which rank@1-5 retrieval results are provided. The correct results are highlighted in green while the incorrect ones are highlighted in red. Best viewed in zoom.

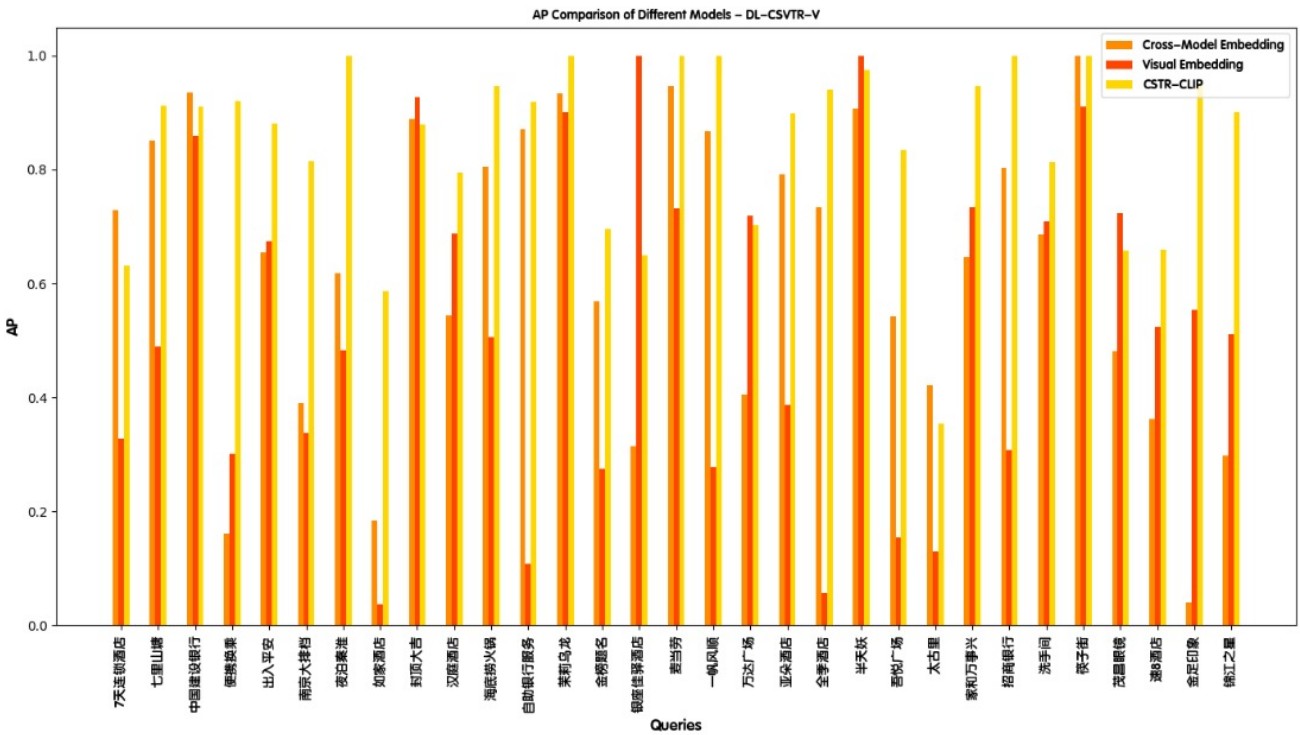

*Figure 12.* Comparison of AP of each query word under the DL-CSVTR-V benchmark.

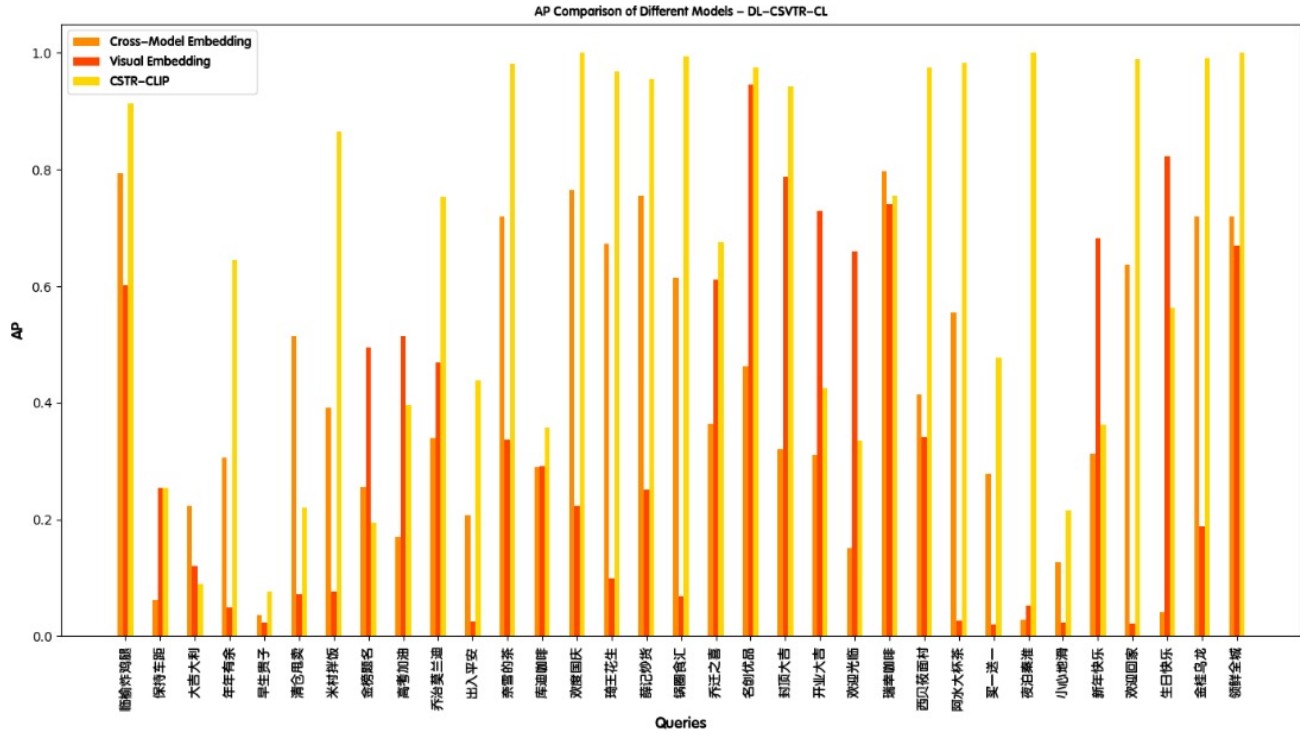

*Figure 13.* Comparison of AP of each query word under the DL-CSVTR-CL benchmark.

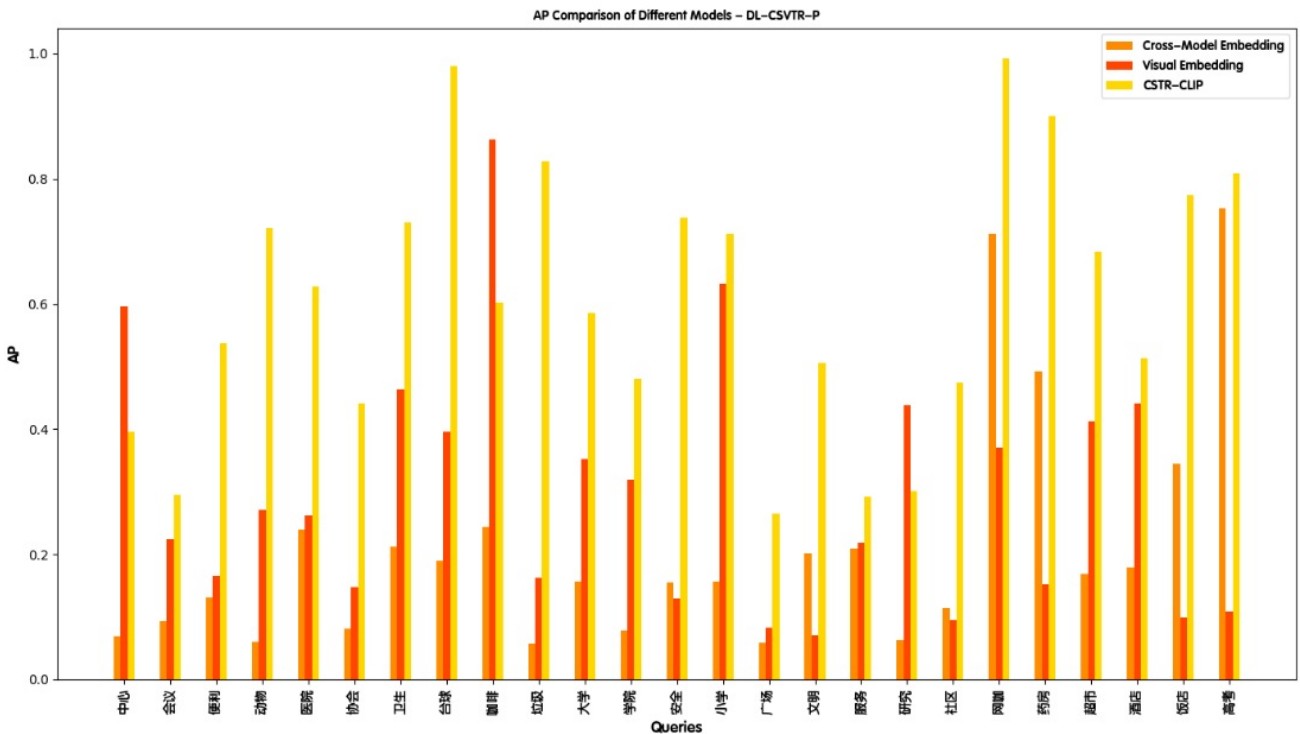

*Figure 14.* Comparison of AP of each query word under the DL-CSVTR-P benchmark.

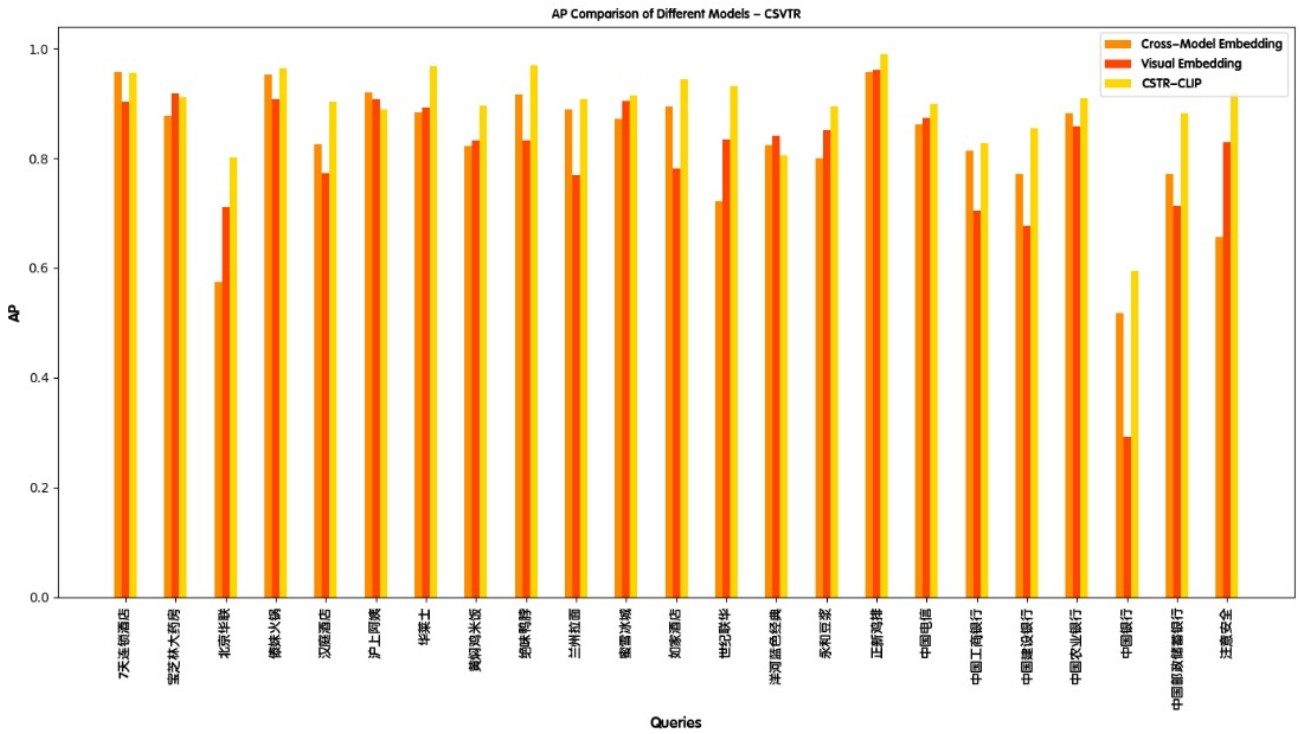

*Figure 15.* Comparison of AP of each query word under the CSVTR benchmark.

