# OpenReview forum: "Beyond Cropped Regions: New Benchmark and Corresponding Baseline for Chinese Scene Text Retrieval in Diverse Layouts"
_ICML.cc/2025/Conference — ICML 2025 poster_

### Official Review · Reviewer_7i8U · 2025-03-09

**Overall Recommendation:** 3

**Summary:**

The paper addresses Chinese scene text retrieval challenges, focusing on the complex layouts of Chinese text in real-world scenes. Current approaches that adapt English text retrieval methods to Chinese contexts show limited performance. The authors introduce DL-CSVTR, a benchmark for evaluating Chinese text retrieval across diverse layouts including vertical, cross-line, and partial alignments --- addressing limitations in existing datasets that primarily feature horizontal text. They also propose CSTR-CLIP, a novel model that moves beyond cropped text regions by employing a two-stage training approach. A key innovation is the Random Alignment Granularity Processing module that improves perception of text elements both within and around text regions. Experiments show CSTR-CLIP outperforms previous methods on both existing benchmarks and the new DL-CSVTR benchmark, particularly for challenging text arrangements.

**Claims And Evidence:**

The paper presents convincing evidence for some claims but falls short in others. The performance improvements of CSTR-CLIP are well-documented in Tables 1 and 2, and the ablation studies in Table 3 effectively demonstrate component contributions.

I find the construction and validation of the DL-CSVTR benchmark problematic. The paper mentions three annotators collected the data but provides minimal details on annotation protocols or quality assurance. For a benchmark paper, this is a significant weakness. Were there inter-annotator agreement measurements? What specific criteria determined layout categories? The dataset size (2,070 images) seems modest for a benchmark intended to evaluate real-world performance.

The causal analysis linking model components to performance gains is weak. For instance, when discussing the RAGP module, the paper shows it helps in cross-line and partial layouts (Table 3, rows 5 vs 6), but doesn't adequately explain the mechanism. Why does random granularity alignment specifically help with these layouts? The visualization in Figure 6 is interesting but doesn't fully connect to quantitative results.

The paper lacks error analysis - when does CSTR-CLIP fail and why? The performance on individual queries varies dramatically (visible in Figures 12-14 in the supplementary material), but this variation isn't analyzed in the main text. Without understanding failure modes, it's difficult to fully assess the model's robustness.

**Essential References Not Discussed:**

Based on my knowledge, I did not identify any major omissions in the related work.

**Experimental Designs Or Analyses:**

I examined several aspects of the experimental design and found some methodological issues:

The DL-CSVTR benchmark construction lacks statistical rigor. The authors use three annotators, but don't report inter-annotator agreement or formalized criteria for categorizing layouts. This raises questions about the benchmark's reliability and reproducibility.

For evaluation metrics, they rely solely on mean Average Precision (mAP), which is standard but insufficient alone. Given their focus on diverse text layouts, layout-specific metrics capturing the unique challenges of vertical or cross-line text would strengthen their analysis.

The ablation study (Table 3) is reasonably designed to isolate component contributions, but lacks error bars or significance testing. With performance differences sometimes being modest (like 88.41% vs 88.57% on CSVTR), it's hard to assess whether improvements are meaningful or statistical noise.

The visualization analysis (Figure 6) offers qualitative insights but feels cherry-picked. A more systematic visualization approach across different query types would better support claims about the model's perceptual abilities.

The baseline implementations warrant scrutiny. The authors replace backbones in previous methods with CLIP for "fair comparison," but this modification fundamentally changes those methods. I question whether these are still valid representations of the original approaches or essentially new hybrid models.

**Methods And Evaluation Criteria:**

The proposed methods align well with the unique challenges of Chinese scene text retrieval. Moving beyond cropped regions is particularly sensible, as Chinese characters often appear in complex spatial arrangements that traditional bounding box approaches can't handle effectively. The two-stage training process reflects a thoughtful consideration of what's actually needed: both OCR capabilities and layout understanding.

I'm less convinced about some aspects of the evaluation framework. While creating a dedicated benchmark for diverse layouts addresses a real gap, the construction feels somewhat ad hoc. The authors manually collected images for different layout categories, but provided little justification for why these specific 89 query terms were chosen or how well they represent real-world retrieval scenarios. A more systematic approach might have started with an analysis of query distributions from actual user data.

The benchmark's size (around 2,000 images) strikes me as minimal. For comparison, standard object detection benchmarks typically include tens of thousands of images. This raises questions about whether performance gains would generalize to larger, more diverse datasets.

One methodological strength is the comparison with both CLIP baselines and prior specialized approaches. This shows the contribution beyond just leveraging a strong pretrained model. The speed-accuracy tradeoffs are also reasonably explored, though real-world deployability considerations could have been discussed more thoroughly.

In summary, while the methods are well-matched to the problem, the evaluation criteria would benefit from more rigorous benchmark construction and validation.

**Other Comments Or Suggestions:**

1. Section 4 describes $Conv _ {Textpos}$ but doesn't specify its architecture details (kernel size, etc.).

2. The explanation of hyperparameters α, β, and θ in RAGP (page 5) lacks justification for chosen values. How sensitive is performance to these settings?

3. The discussion of inference speed (FPS) in results sections mentions improvements but doesn't analyze computational complexity of different components.

**Other Strengths And Weaknesses:**

### Strengths:
1. The paper addresses a practical problem in Chinese scene text retrieval that differs significantly from English text retrieval challenges. The authors effectively identify limitations in applying English-centric methods to Chinese texts with diverse layouts.

2. Their technical approach cleverly combines CLIP's visual understanding with layout-specific guidance. Rather than merely cropping text regions, they preserve global context while directing attention to text areas.

3. The two-stage training process shows careful consideration of the problem's unique requirements. The first stage focuses on core OCR abilities while the second enables handling complex layouts.

### Weaknesses:
1. The experimental analysis lacks depth in connecting model behavior to performance across different layout types. While ablation studies show component contributions, they don't sufficiently explain why certain components help specific layouts.

2. The performance differences shown in supplementary figures (12-15) reveal considerable variation across query types that deserves more thorough analysis. This would provide more meaningful insights than the aggregate metrics alone.

3. Several technical descriptions need clarification, particularly for the Random Alignment Granularity Processing. The algorithm is central to their approach but described in abstract terms without concrete examples of how it transforms inputs.

4. The paper would benefit from more direct comparison with recent transformer-based approaches that handle complex layouts, situating their work in the broader context of both retrieval and OCR research.

**Questions For Authors:**

N/A

**Relation To Broader Scientific Literature:**

The paper makes two significant contributions that connect to several research threads:

The DL-CSVTR benchmark extends previous scene text retrieval datasets (like IIIT and CSVTR), addressing their notable bias toward horizontal text layouts. While prior benchmarks established evaluation frameworks, they failed to capture layout diversity in real Chinese text.

CSTR-CLIP relates to recent advances in guided attention for vision-language models. Similar to MaskCLIP and Red-Circle's approach to regional attention, it directs CLIP's focus while preserving global context. However, its full-image approach represents a departure from the dominant cropped-region paradigm established in Mishra's 2013 work and continued through Gomez (2018) and Wang (2021).

The multi-granularity alignment concept evolved from cross-modal embedding work by Luo and Wen, but introduces flexibility previous approaches lacked.

A notable gap is limited engagement with transformer-based OCR systems that have shown promise for complex layouts. By positioning their work primarily within retrieval rather than OCR literature, the authors somewhat constrain their conceptual framework.

**Theoretical Claims:**

The paper is primarily empirical in nature and does not present formal mathematical proofs or theoretical guarantees that require verification. The contributions are algorithmic and experimental rather than theoretical.

The authors do make some informal claims about why their approach works, particularly regarding the limitations of cropped-region paradigms and the benefits of multi-granularity alignment, but these are supported through ablation studies and experimental results rather than formal proofs.

The paper's technical foundation largely builds on the existing CLIP architecture with modifications specific to the Chinese scene text retrieval task. The authors explain their algorithmic contributions (like the Random Alignment Granularity Processing module) but don't provide theoretical convergence guarantees or complexity analyses that would typically accompany theoretical papers.

Given the applied nature of the work, the absence of formal proofs is not necessarily a weakness. However, a stronger theoretical analysis of why multi-granularity alignment specifically addresses the challenges of diverse Chinese text layouts could have strengthened the paper's contributions beyond empirical results.

---

> ### Author Rebuttal · Authors · 2025-03-28
>
> Thank you for your comment, and we would like to clarify these questions according to subjects.
>
> **Question about DL-CSVTR datasets**
>
> 1. Claims And Evidence's Para 2
>
>    We ensured that the process involved three annotators, with one main annotator overseeing the data quality and consistency. The main annotator was responsible for validating the layout classification, image quality, and privacy considerations of the data submitted by the other two annotators. Additionally, the data provided by the main annotator was shared with the other annotators for reference. After annotating six query words, all three annotators would engage in discussions to identify and resolve any inconsistencies, thus ensuring alignment in the annotations. Regarding layout categories, we employed manual quality control to ensure that the visual representation of query words in images strictly adhered to the corresponding layout categories. We ensured that no additional layout formats for the same query word interfered with the classification.
>
> 2. Methods And Evaluation Criteria's Para 3
>
>    DL-CSVTR refers to the scale of scene text retrieval benchmarks in both Chinese and English scene text retrieval tasks, such as CSVTR, IIITSceneTextRetrieval, and StreetViewText. This approach ensures that the dataset is appropriately aligned with existing benchmarks, making it well-suited for evaluating scene text retrieval performance.
>
> 3. Methods And Evaluation Criteria's Para 2
>
>    The 89 query words, sourced from real street scenes including trademarks and phrases, were manually selected and clustered by annotators based on layout requirements in large-scale images. Data collection preceded model design, as our prior research identified gaps in handling specific Chinese text layouts. To address this, we created DL-CSVTR using the CSVTR methodology, ensuring the dataset's relevance and authenticity with real-world street view scenes.
>
> **Question about CSTR-CLIP model**
>
> 1. Weaknesses 1, 3 & Experimental Designs Or Analyses's Para 3
>
>    RAGP relaxes the strict constraints of location-based suggestions. After the first training stage, the model matches regions indicated by Textpos Conv. However, layouts like cross-line and partial layouts may only appear partially within the detection region. RAGP mitigates this by adjusting the alignment of the suggestion region with the query, improving performance for layouts that don’t fully align, especially for cross-line and partial layouts.
>
>    The modest improvement with CSVTR is because RAGP addresses location constraints, which are less relevant for horizontally aligned query words, as shown in Figure 2.
>
>    We also appreciate the reviewer’s suggestion to include the pseudocode for RAGP. We will provide a more detailed description of the RAGP algorithm in the camera-ready paper.
>
> 2. Other Comments Or Suggestions 1
>
>    we adopted the same kernel size as the first convolutional layer of CLIP's image data preprocessing to maintain compatibility with subsequent fusion.
>
> 3. Other Comments Or Suggestions 2
>
>    The variation in retrieval accuracy across different query words is most evident in cross-line and partial layouts, due to differences in the spatial distances between query word parts in cross-line layouts and the varying paragraph lengths in partial layouts. These issues are closely related to RAGP, which guides the model to focus on the visual features around the suggested region. The effectiveness of RAGP is influenced by hyperparameters α, β, and θ, which control the receptive field size. We conducted experiments and visualized bad cases to determine the optimal combination of these parameters during training.
>
> **Question about experiment design**
>
> 1. Weaknesses 4
>
>    We chose GOT as a transformer-based approach, using spotting-related instructions. Specifically, edit distances between a query word and the spotted words from scene images are used for text-based image retrieval.
>
>    | Method |  CSVTR | DL-CSVTR-V |  DL-CSVTR-CL |  DL-CSVTR-P |
>    |-----|-----|-----|-----|-----|
>    | GOT |86.56| ***84.91*** |  59.47 | 55.78 |
>    | Ours |***88.57***| 84.44 | ***65.56*** | ***61.83*** |
>
> 2. Experimental Designs Or Analyses Para 3
>
>    Recall is suitable for single-target queries, but in DL-CSVTR, a query word can correspond to multiple targets, making it unsuitable. mAP, which has been used as the sole metric for accuracy in previous scene text retrieval works, is therefore employed in this study.
>
> 3. Other Comments Or Suggestions 3
>
>    Please refer to Reviewer QR6X's response 1.
>
> 4. Weaknesses 2
>
>    Errors mainly occur in cross-line layouts with large gaps between query word parts and in partial layouts where the query word occupies a small portion of the string. These issues contribute to bias in errors. RAGP with global feature fusion improved performance, but for challenging samples, further attention to semantically related elements outside the suggested region may be needed.

---

> > ### Comment · Reviewer_7i8U · 2025-04-07
> >
> > The rebuttal provides satisfactory answers to several of my concerns, and the authors seem receptive to feedback. While this is encouraging, I still believe the current version of the paper does not fully meet the threshold for a higher score. Hence, I am keeping my original evaluation.

---

### Official Review · Reviewer_QR6X · 2025-03-13

**Overall Recommendation:** 4

**Summary:**

This paper focuses on Chinese scene text retrieval, which aims to extend previous English scene text retrieval to Chinese. The authors establish a Diversified Layout benchmark for Chinese Street View Text Retrieval (DL-CSVTR) to assess retrieval performance across different text layouts. They also propose Chinese Scene Text Retrieval CLIP (CSTR-CLIP), a new model integrating global visual information and multi-granularity alignment training. Experiments on existing benchmarks show that CSTR-CLIP achieves an 18.82% accuracy improvement over the previous SOTA model and has a faster inference speed. Analysis of DL-CSVTR validates its superiority in handling diverse text layouts.

**Claims And Evidence:**

Yes, confirmed.

**Essential References Not Discussed:**

I hold there are relatively sufficient essential references discussed.

**Experimental Designs Or Analyses:**

Yes. I have checked all the experimental results, including the comparison with previous SOTA and the ablation of modules of all stages. The experiments are sound and extensive and are superior to previous works.

**Methods And Evaluation Criteria:**

Yes, confirmed.

**Other Comments Or Suggestions:**

Please refer to the above comments.

**Other Strengths And Weaknesses:**

Strengths:
1) This paper constructs a Chinese scene text retrieval dataset, which includes various layouts, and can support vertical/cross-line/partial text retrieval that English text retrieval seldom encounters.
2) They propose CSTR-CLIP method for Chinese scene text retrieval, which relieves text detection needs and enhances perception flexibility through multi-granularity alignment.
3) Extensive experiments show that the proposed method achieves superior performance on both the previous Chinese scene text retrieval dataset and the proposed DL-CSVTR dataset.

Weaknesses:
1) Tab.1, why CSTR-CLIP is much faster than all other methods except CLIP should be explained in detail.
2) Fig.7, what is the application for “Interactive Region-Specified Scene Text Retrieval”? If the user can give the mask, why do they not only input the mask region as a separate image?

**Questions For Authors:**

Please refer to the above comments.

**Relation To Broader Scientific Literature:**

Extending text retrieval task to Chinese scenario. It could identify limitations in prior research.

**Theoretical Claims:**

This submission does not involve proof of theory.

---

> ### Author Rebuttal · Authors · 2025-03-26
>
> **Question 1**
>
> Tab.1, why is CSTR-CLIP much faster than all other methods except CLIP? This should be explained in detail.
>
> **Response 1**
>
> Thank you for the valuable comment. The faster performance of CSTR-CLIP compared to other methods can be attributed to the simplified nature of our approach. Previous methods based on visual embedding techniques such as [1][2], require additional computational effort to crop text regions, and apply model-based style processing to convert those regions into a standardized format for easier matching. Furthermore, these methods also required extra steps to transform the query into an image format that could match the cropped text regions, leading to a significant loss in inference speed. Additionally, earlier approaches based on cross-modal embeddings [3] involved extra costs for designing matching templates and performing region cropping, which further slowed down the inference process.
>
> In contrast, CSTR-CLIP benefits from a direct cross-modal matching approach, eliminating the need for extra style transformations, rendering, or template-based matching constraints. This streamlined process allows for much faster inference speed. We will clarify this explanation in the manuscript to provide a deeper understanding of the performance improvements.
>
> **Question 2**
>
> Fig.7, what is the application for "Interactive Region-Specified Scene Text Retrieval"? If the user can give the mask, why do they not only input the mask region as a separate image?
>
> **Response 2**
>
> We appreciate your thoughtful question. "Interactive Region-Specified Scene Text Retrieval" is designed to refine the granularity of user searches. When users need to search for images containing specific text within their image library and have some recollection of the region where the text appears, they can provide a region suggestion. By focusing on the region of interest, this enhances the search results, improving accuracy and recall in retrieval tasks. However, since the region specified by the user may not perfectly match, directly cropping the image could lead to information loss and additional computational costs. Thanks to the design and training paradigm of CSTR-CLIP, the model does not limit itself to the given region but also responds to surrounding areas within the suggested region, as illustrated in Figure 6. Therefore, CSTR-CLIP can improve retrieval accuracy and recall by considering the user's region suggestion and query text, all while avoiding information loss.
>
> [1]Visual and semantic guided scene text retrieval
>
> [2]Visual Matching is Enough for Scene Text Retrieval
>
> [3] Scene text retrieval via joint text detection and similarity learning.

---

### Official Review · Reviewer_UNMy · 2025-03-14

**Overall Recommendation:** 3

**Summary:**

This paper addresses the limitations of existing Chinese scene text retrieval methods, which inherit the solution for English scene text retrieval and fail to achieve satisfactory performance in Chinese scene text retrieval. Therefore, the authors first introduce DL-CSVTR, a new benchmark featuring vertical, cross-line, and partial text layouts for more realistic assessments. Then, they propose CSTR-CLIP. CSTR-CLIP applies a two-stage training process to overcome previous limitations, such as the exclusion of visual features outside the text region and reliance on single-granularity alignment, thereby enabling the model to effectively handle diverse text layouts. Experimental results show that CSTR-CLIP outperforms existing methods significantly on both the standard CSVTR dataset and the new DL-CSVTR benchmark, effectively handling varied text layouts.

**Claims And Evidence:**

Yes, I think the claims in this submission are supported by their experiments and discussions.

**Essential References Not Discussed:**

I think this paper adequately cites and discusses the essential related works that are necessary for understanding the context of its key contributions.

**Experimental Designs Or Analyses:**

Yes, I reviewed the experimental design and related analyses and found them reasonable and valid.

**Methods And Evaluation Criteria:**

Yes, I think the proposed method and evaluation criteria are proper and make sense for the problem.

**Other Comments Or Suggestions:**

I have no more extra comments.

**Other Strengths And Weaknesses:**

Strengths:
- It is nice to see this work proposes a new dataset DL-CSVTR. This dataset is interesting and contains a large number of challenges from real-world scenarios.
- Performance of this work is quite well. Especially in those challenge cases such as vertical, cross-line, and partial alignments.

**Questions For Authors:**

1. In Table 2, we found that CSTR-CLIP has significantly improved performance in various challenge scenarios. However, the basic horizontal cases are not reported together. Can the author provide the provide the corresponding performance results?

2. Although this work is designed for Chinese scene text retrieval. However, all the problems mentioned are also challenges for the English scene text retrieval task. Therefore, can the author provide the performance results of CSTR-CLIP on English scene text retrieval datasets? I think it will be helpful to evaluate its improvement upon recent works on more general datasets.

**Relation To Broader Scientific Literature:**

This work focuses on Chinese scene text retrieval and proposes complex and variable datasets. Therefore, I think it makes sense for a border impact in the whole scene text retrieval community. Besides the dataset, the author also introduces a method named CSTR-CLIP, which uses full image information with guided attention instead of previous crop-detection processing. I think it is somewhat reasonable and interesting.

**Theoretical Claims:**

Yes, I have checked the correctness of the proofs for the theoretical claims and found no issues.

---

> ### Author Rebuttal · Authors · 2025-03-26
>
> **Question 1**
>
> In Table 2, we found that CSTR-CLIP has significantly improved performance in various challenge scenarios. However, the basic horizontal cases are not reported together. Can the author provide the corresponding performance results?
>
> **Response 1**
>
> We appreciate the reviewer’s comment. We have taken CSVTR as the baseline for horizontal retrieval, as most query text in the dataset (as shown in Figure 2) predominantly follows a horizontal layout. Therefore, the performance of CSTR-CLIP in horizontal retrieval scenarios is essentially reflected in the CSVTR baseline results. For further reference, the results in Table 1 of the paper can provide additional insights into the performance for horizontal retrieval.
>
> To elaborate, we manually categorized the visual representations of the query text in the CSVTR dataset based on their corresponding layouts. In cases where a single image contained multiple orientations, such as both horizontal and vertical text, we applied a priority-based classification process. Specifically, if both orientations were fully visible and intact, we followed this priority: cross-line > partial > vertical > horizontal. For instance, if both horizontal and vertical layouts appeared in the same image, we classified the layout as vertical. After conducting this analysis, we found that 92.62% of the data had query words with a horizontal visual representation in the image. Consequently, we have used CSVTR as the baseline for horizontal layout retrieval and did not collect additional horizontal layout data in the DL-CSVTR dataset.
>
> **Question 2**
>
> Although this work is designed for Chinese scene text retrieval, all the problems mentioned are also challenges for the English scene text retrieval task. Can the author provide the performance results of CSTR-CLIP on English scene text retrieval datasets? I think it will be helpful to evaluate its improvement upon recent works on more general datasets.
>
> **Response 2**
>
> Thank you for raising this important point. We agree that the challenges discussed in our work are indeed also relevant to English scene text retrieval. In response, we trained CSTR-CLIP using the en-clip model with the same processing approach and evaluated its performance on English scene text retrieval datasets.
>
> Regarding the data, we followed the methodology used in SynthText-900K for the English corpus and generated a dataset of 300k images, matching the scale of the pre-trained data used for the Chinese scene text retrieval. For real-world data in the second stage, we used the English subset of the MLT dataset as the training set, aligning it with previous work in English scene text retrieval.
>
> In terms of model design, we retained the two-stage framework of CSTR-CLIP, replacing the cn-clip with en-clip as the initialization model. We chose the IIIT Scene Text Retrieval dataset as the benchmark for English scene text retrieval, using mean Average Precision (mAP) as the evaluation metric to compare our method's performance in English scene text retrieval. The results are shown in the table below:
>
> | **Method**          | **mAP** |
> |-------------------------------|---------|
> | TDSL                | 77.09   |
> | VSTR                | 77.40   |
> | Luo et al           | 82.15   |
> | Ours (Stage 1)      | 83.17   |
> | Ours (Stage 2)      | ***86.75***   |
>
>
> CSTR-CLIP also demonstrated excellent performance in the English scene text retrieval context. However, we observed that cross-line and partial layouts are less common in the English scene. This was evident in the visual representation of query words in the IIIT Scene Text Retrieval dataset, which showed a similar trend. We believe the key to performance improvement lies in the retention of full-image information and the guidance provided by region suggestions. This is particularly important because many images in the IIIT dataset have lower clarity, and previous crop-based methods struggle to effectively retrieve text from such images. The second-stage improvement likely comes from fine-tuning on real-world data. Therefore, the "Beyond Cropped Regions" paradigm is indeed beneficial for English scene text retrieval as well. We also plan to open-source CSTR-CLIP for English scene text retrieval in the near future.

---

> > ### Comment · Reviewer_UNMy · 2025-04-07
> >
> > Thanks for the response from the author. I have no more questions about it. It is a well-organized paper. The advantage of this work is that the motivation is straightforward and easy to follow. The method is reasonable and highly related to its motivation. The disadvantage of this work is that the proposed method is somewhat simple. Overall, despite some minor flaws, the strengths of this paper outweigh its weaknesses, and I am inclined to maintain my weak accept recommendation.

---

### Official Review · Reviewer_osp1 · 2025-03-14

**Overall Recommendation:** 4

**Summary:**

In this paper, the authors aim to solve the problem of Chinese scene text retrieval in complex and diverse layouts. They first establish the DL-CSVTR benchmark including vertical, cross-line and partial alignments. In addition, the authors propose CSTR-CLIP method which integrates global visual information with multi-granularity alignment training. The experiments are conducted on both previous benchmark and the proposed DL-CSVTR demonstrating the proposed method CSTR-CLIP outperforms previous SOTA model.

## update after rebuttal
I maintain my score after reading rebuttal

**Claims And Evidence:**

The claims made in this paper include the specific characteristics of Chinese scene text retrieval, which is evident since Chinese is different from English obviously.

**Essential References Not Discussed:**

NO

**Experimental Designs Or Analyses:**

The performance on CSVTR benchmark, DL-CSVTR benchmark and ablation study are all checked, and the results sound solid.

**Methods And Evaluation Criteria:**

They are reasonable because Chinese has more vertical textlines than English, and cross-line and partial retrieval are useful because Chinese textline is composed of characters while English textline is composed of words.

**Other Comments Or Suggestions:**

see the weakness part.

**Other Strengths And Weaknesses:**

Strengths:

(1) In the paper, the authors find that Chinese scene text retrieval is different from English in that Chinese text has many layouts, so they do not only transfer English methods to Chinese scenario but design new retrieval tasks including vertical, cross-line and partial retrieval where English retrieval seldom involves. This is significant since Chinese text together with other similar texts are also used all over this world.

(2) The proposed CSTR-CLIP method further pushes the CLIP into Chinese scene text retrieval task, verifying that CLIP has the potential in this task though it is the by-product. The CSTR-CLIP performs well on Chinese benchmarks.

(3) The experiments are enough to verify the effectiveness of the proposed method, and many extra experimental results are supplied in the appendix material.

Weaknesses:

(1) In figure 2, we can conclude that Horizontal text occupies 92.62%. I is not clear about how is the oriented text classified? To horizontal or vertical?

(2) Other than vertical, cross-line and partial text retrieval, is there any other type of retrieval task that has not been well addressed?

**Questions For Authors:**

see the weakness part.

**Relation To Broader Scientific Literature:**

t pushes the general CLIP model to extensive text retrieval task with sophisticated design.

**Theoretical Claims:**

It seems there is no proof for theoretical claim involved.

---

> ### Author Rebuttal · Authors · 2025-03-26
>
> **Question 1**
>
> In Figure 2, we can conclude that horizontal text occupies 92.62%. It is not clear how the oriented text is classified – is it classified as horizontal or vertical?
>
> **Response 1**
>
> We appreciate the reviewer’s observation. To clarify, we classified the visual representation of query text in the CSVTR dataset based on manual verification of their layout in the images. Specifically, when both horizontal and vertical text appear in the same image, we categorize the query text based on the priority of visibility and completeness. The priority order for classification is as follows: cross-line > partial > vertical > horizontal. In other words, if both horizontal and vertical text are present in the same image, the text would be classified as vertical. This approach ensures a consistent and logical classification across all images, accounting for different layout variations.
>
> **Question 2**
>
> Other than vertical, cross-line, and partial text retrieval, is there any other type of retrieval task that has not been well addressed?
>
> **Response 2**
>
> In terms of Chinese text layouts, there is another layout type that we believe requires further attention: dispersed layout. For instance, a query term like "ICML" could appear in an image as "International Conference on Machine Learning", where the characters are spaced out. This scenario represents a potential area of improvement, and we are working on solutions to address this issue.
>
> Furthermore, we believe two additional challenges in scene text retrieval should be explored in future work: 1) Retrieval of text with specific attributes, such as layout and color; 2) Retrieval of text in relation to visual elements, for example, "a blue building with the text 'xxx' on it." We see these as promising directions for future advancements in scene text retrieval. We will discuss it in the future work in our camera-ready version.

---

### Decision · Program_Chairs · 2025-05-01

**Decision:**

Accept (poster)

**Comment:**

All four reviewers recommend accepting this paper after rebuttal and the final scores of reviewers are 2 Weak accept and 2 Accept. The AC agrees with the reviewers that the paper has made some valuable contributions (i.e., a new benchmark and a new baseline for Chinee scene text retrieval) to the community and deserves to be published in ICML.